# Chondroitin Sulfate and Fucosylated Chondroitin Sulfate as Stimulators of Hematopoiesis in Cyclophosphamide-Induced Mice

**DOI:** 10.3390/ph14111074

**Published:** 2021-10-24

**Authors:** Nadezhda E. Ustyuzhanina, Natalia Yu. Anisimova, Maria I. Bilan, Fedor V. Donenko, Galina E. Morozevich, Dmitriy V. Yashunskiy, Anatolii I. Usov, Nara G. Siminyan, Kirill I. Kirgisov, Svetlana R. Varfolomeeva, Mikhail V. Kiselevskiy, Nikolay E. Nifantiev

**Affiliations:** 1N.D. Zelinsky Institute of Organic Chemistry, Russian Academy of Sciences, Leninsky Prospect 47, 119991 Moscow, Russia; bilan@ioc.ac.ru (M.I.B.); usov@ioc.ac.ru (A.I.U.); 2N.N. Blokhin National Medical Research Center of Oncology, Kashirskoe Shosse 24, 115478 Moscow, Russia; n_anisimova@list.ru (N.Y.A.); donenko.f20010@yandex.ru (F.V.D.); nara19922@yandex.ru (N.G.S.); kirgiz-off@yandex.ru (K.I.K.); varfolomeeva-07@mail.ru (S.R.V.); 3V.N. Orekhovich Research Institute of Biomedical Chemistry, Pogodinskaya Str. 10, 119121 Moscow, Russia; galina.morozevich@ibmc.msk.ru (G.E.M.); yashunsky1959@yandex.ru (D.V.Y.)

**Keywords:** graft versus host disease, granulocyte colony-stimulating factor, chondroitin sulfate, fucosylated chondroitin sulfate, hematopoiesis, immunosuppression, cyclophosphamide

## Abstract

The immunosuppression and inhibition of hematopoiesis are considered to be reasons for the development of complications after intensive chemotherapy and allogeneic hematopoietic stem cell transplantation. Chondroitin sulfate (**CS**), isolated from the fish *Salmo salar*, and fucosylated chondroitin sulfate (**FCS**), isolated from the sea cucumber *Apostichopus japonicus*, were studied for their roles as stimulators of hematopoiesis in a model of cyclophosphamide-induced immunosuppression in mice. The recombinant protein r G-CSF was applied as a reference. The studied polysaccharides were shown to stimulate the release of white and red blood cells, as well as platelets from bone marrow in immunosuppressed mice, while r G-CSF was only responsible for the significant increase in the level of leucocytes. The analysis of different populations of leucocytes in blood indicated that r G-CSF mainly stimulated the production of neutrophils, whereas in the cases of the studied saccharides, increases in the levels of monocytes, lymphocytes and neutrophils were observed. The normalization of the level of the pro-inflammatory cytokine IL-6 in the serum and the recovery of cell populations in the spleen were observed in immunosuppressed mice following treatment with the polysaccharides. An increase in the proliferative activity of hematopoietic cells CD34(+)CD45(+) was observed following ex vivo polysaccharide exposure. Further study on related oligosaccharides regarding their potential as promising drugs in the complex prophylaxis and therapy of hematopoiesis inhibition after intensive chemotherapy and allogeneic hematopoietic stem cell transplantation seems to be warranted.

## 1. Introduction

In spite of its numerous side effects, the use of chemotherapy in cancer treatment is still regarded as the main method of fighting the disease [1,2,3]. Cytostatics such as cyclophosphamide (CPh), cisplatin (Cis) and doxorubicin (DOX), as well as the toxic effects they display on cancer cells, also suppress the proliferation of blood cells in bone marrow [4,5], lead to the desolation of the spleen [6,7] and cause damage to myocardial and hepatic cells [7,8,9,10,11,12]. These effects result in severe complications, including acute neutropenia, lymphopenia, erythropenia, thrombocytopenia, hepatic fibrosis and cirrhosis and different cardiovascular diseases.

Allogeneic hematopoietic stem cell transplantation (allo-HSCT), used in the treatment of many hematological disorders, is still limited by severe complications and transplant-related mortality (TRM) [13,14]. Acute graft versus host disease (aGvHD) is the leading cause of morbidity and TRM following allo-HSCT. For the prevention of TRM, immunosuppressants are used, including cyclophosphamide, which leads to the inhibition of hematopoietic recovery [15,16].

The aforementioned inhibition of hematopoiesis is often achieved using therapeutic and prophylactic treatments. Recombinant granulocyte colony-stimulating factor (r G-CSF) is widely used in medical practice to treat patients who have neutropenia [17]. However, r G-CSF does not stimulate other hematopoietic germs. Therefore, there is high demand for the development of a drug capable of increasing the levels of all types of blood cells for use in the supportive therapy of cancer patients.

Recently, the sulfated polysaccharides fucoidan (**CF**), isolated from the brown seaweed *Chordaria flagelliformis*, and fucosylated chondroitin sulfate (**MM**), isolated from the sea cucumber *Massinium magnum*, were found to be effective stimulators of hematopoiesis in a model of cyclophosphamide-induced immunosuppression in mice [18]. Similarly to r G-CSF, these polysaccharides effectively stimulated neutropoiesis, and additionally, they were shown to be capable of stimulating erythropoiesis and thrombocytopoiesis. Fucoidan (**CF**) is built of repeating (1→3)-linked α-L-fucopyranosyl residues, some of which bear α-D-glucuronyl and more complex oligosaccharide branches [19]. The random sulfation of a backbone and branches significantly masks the regularity of the polysaccharide. Fucosylated chondroitin sulfate (**MM**) consists of the chondroitin core [→4)-β-D-GlcA-(1→3)-β-D-GalNAc-(1→]_n_ decorated by 3,4-di-O-sulfated α-L-fucosyl branches attached to O-3 of GlcA units and by sulfates at O-4 and/or O-6 of GalNAc [20]. Chemically modified fucoidan and related linear oligosaccharides have also demonstrated the ability to stimulate hematopoiesis [21].

In this paper, we describe the results of the ongoing study of structurally different sulfated polysaccharides as stimulators of hematopoiesis and report the results from the investigation of the linear chondroitin sulfate (**CS**) [22] from the fish *Salmo salar* and the branched fucosylated chondroitin sulfate (**FCS**) [23] from the sea cucumber *Apostichopus japonicus* (Figure 1).

## 2. Results

The polysaccharide sample **CS** was prepared from a crude extract of *Salmo salar* cartilage via mild alkaline treatment, followed by anion-exchange chromatography, as described previously [22]. This biopolymer is made up of a linear chondroitin core [→3)-β-D-GalNAc-(1→4)-β-D-GlcA-(1→]_n_ sulfated at O-4 or at O-6 of GalNAc and thus related to chondroitin sulfates A and C, respectively (Figure 1). The ratio of **A** to **C** disaccharide blocks was about 0.8. The molecular weight of **CS** was estimated to be 9 kDa.

The polysaccharide sample **FCS** was isolated from a body wall of the sea cucumber *Apostichopus japonicus*, as described previously [23]. This biopolymer contains a typical chondroitin core [→3)-β-D-GalNAc-(1→4)-β-D-GlcA-(1→]_m_, bearing sulfate groups at O-4 or both at O-4 and O-6 of GalNAc, as well as α-L-Fuc branches attached to O-3 of GlcA units (Figure 1). Notably, the fucosyl branches were different with respect to their patterns of sulfation: Fuc2*S*4*S*, Fuc3*S*4*S* and Fuc4*S* were found in a ratio of about 6:2:1. The molecular weight of **FCS** was estimated to be 27 kDa.

Previously, we showed that sulfated polysaccharides did not affect hematological parameters in intact animals [18,22]. In this work, we describe the effects of **CS** and **FCS** on the release of blood cells from bone marrow on a model of CPh-induced immunosuppression in mice. Recombinant protein r G-CSF (Leicyta) was applied as a reference. Intact animals were regarded as a positive control. The non-toxic dosage regime of the compounds and the mode of the experiment have previously been elaborated upon [18,21,22]. The number of white and red blood cells (WBCs, RBCs) and the number of platelets were determined. The values of the hematological parameters in various groups of mice are presented in Figure 2.

Analysis of the results indicated that the administration of CP to mice was accompanied by a sharp drop in the WBC blood concentration, as well as a moderate decrease in the concentrations of RBCs and platelets. The subsequent administration of r G-CSF to these animals contributed to a significant increase in the number of WBCs, which even exceeded the value in the control group, but did not affect the concentrations of RBCs and platelets. On the contrary, a tendency toward thrombocytopenia was observed with the use of this drug. Quite a different picture was observed after the treatment of mice with CPh-induced immunosuppression with polysaccharides **CS** and **FCS**, which led to a restoration of the number of both WBCs and RBCs, as well as platelets, to the control levels in blood.

The detailed examination of WBC populations in the blood of immunosuppressed mice showed that r G-CSF possessed a narrow spectrum of effect on blood cells. In particular, it stimulated an increase in the level of neutrophils in blood, while the level of lymphocytes decreased by up to 80% in comparison to the control values (Figure 3). The effects of **CS** and **FCS** were more balanced; an increase in the number of neutrophils was accompanied by an increase in the concentrations of monocytes and lymphocytes.

It is known that an increase in the serum level of this cytokine is connected with damage to the liver parenchyma [8]. Therefore, IL-6 can be considered as not only a sign of hepatotoxicity, which appears to be a frequent complication with the use of anticancer drugs [24,25,26], but also as a sign of pathogenic factors of GVHD. Our data confirmed the results of the previous studies that demonstrated an increasing level of this cytokine in blood serum of mice with CPh-induced immune suppression. At the same time, in comparison with r G-CSF, the use of **CS** and **FCS** contributed to a more pronounced normalization of this parameter than the level in the control. The change in serum IL-6 level in immunosuppressed mice after the administration of all tested compounds was evaluated in comparison with the control levels (Figure 4).

Analysis of spleen morphology in the experimental and control groups of mice was performed, as the spleen is one of the main organs of the immune system (Figure 5). In mice with CPh-induced immunosuppression without additional treatment, a noticeable depletion of the pool of immunocompetent mononuclear cells in the spleen was observed, while the administration of **CS** and **FCS** to animals contributed to the replenishment of a significant portion of the pool of lymphoid cells in the spleen pulp. This was similar to the effect of r G-CSF. Representative pictures of spleen tissues are presented in Appendix A.

Special experiments were carried out in vitro in order to clarify the mechanism of action of **CS** and **FCS**. Bone marrow cells from healthy mice were incubated ex vivo in a nutrient medium containing the cytostatic agent cisplatin (Cis), r G-CSF, **CS**, **FCS** or a combination thereof (Cis-r G-CSF, Cis-**CS** and Cis-**FCS**) for 2 days. Intact cells were used as a control. Cisplatin was chosen as a cytostatic agent because it is a component of most combinations of chemotherapeutic drugs used to treat cancer patients [2]. In addition, unlike CPh, it is capable of exerting a cytotoxic effect without additional transformation in the patient’s body. After 2 days of incubation, the number of living cells and the level of their proliferation were assessed using flow cytometry (Figure 6, Figure 7 and Figure 8). It was shown that similarly to r G-CSF, **CS** and **FCS** increased the number of living cells following cisplatin exposure and stimulated their proliferation (increasing the number of Ki67(+) cells).

The elucidation of the proliferation of the special low numerous population of the hematopoietic cells CD34(+)CD45(+) revealed the same tendency. In the cases of **CS** and **FCS**, we observed an increase in the concentration of the proliferating hematopoietic cells [CD34(+)CD45(+)Ki67(+)] by 8 and 5 times, respectively, after cytostatic agent exposure (Figure 8).

Additionally, the expression of the cell adhesion marker CD44 on the membrane of nuclear bone marrow cells was assessed following treatment with r G-CSF, **CS** and **FCS** (Figure 9). The studied polysaccharides were shown to stimulate CD44 expression, while r G-CSF did not demonstrate any activity in this test.

To study the effect of the polysaccharides **CS** and **FCS** on cell adhesion, bone marrow cells were incubated ex vivo in the presence of these biopolymers and analyzed using an RTCA ACEA cell analyzer. The cell index measured by this method is directly proportional to the density of cell adhesion to the bottom of the well and to the number of the cells. It was shown that **CS** and **FCS** led to an increase in the cell index after 4 hours of incubation (Figure 10). This could be considered as evidence of the polysaccharides’ stimulation of cell adhesion to the bottom of the wells. Representative photos of the bottom of the wells confirmed this conclusion (Appendix A). Indeed, the formation of multicellular clusters was noted in the **CS** and **FCS** groups; in the control group, these structures were fewer and smaller, while there were no clusters in the Cis group. Considering the results of flow cytometry, it can be assumed that the formation of clusters is connected to the increased expression of cell adhesion molecules CD44.

## 3. Discussion

Treating cancer patients with high-dose chemotherapy often leads to hematopoietic dysfunction, which manifests in the form of anemia, myelosuppression and immunosuppression. These side effects hinder the continuation of treatment for patients and, therefore, can decrease the effectiveness of chemotherapy. Recombinant granulocyte colony-stimulating factor (r G-CSF) is most commonly used to treat neutropenia, and recombinant human erythropoietin is used to stimulate erythropoiesis [17,27,28]. Although r G-CSF is used to treat chemotherapy-induced neutropenia, it can also have a stimulatory effect on tumor cells, promoting tumor stem cell longevity and tumor cell proliferation and migration. In addition, it can stimulate the procancerogenic phenotype of immune cells (M2 macrophages, myeloid suppressor cells and regulatory T cells) [29]. The most important adverse events reported with erythropoietin administration are arterial hypertension, cerebral convulsion/hypertensive encephalopathy, thromboembolism, iron deficiency and influenza-like syndrome [30]. In addition, the use of these drugs leads to the stimulation of only one hematopoietic germ (erythropoiesis or neutropoiesis), while in many cases, high-dose chemotherapy leads to pancytopenia and requires the simultaneous stimulation of all components of hematopoiesis.

The issue of the restoration of hematopoietic dysfunction is especially acute in allogeneic hematopoietic stem cell transplantation (HSCT) in children with hematopoietic malignancies. Preparation for HSCT includes high-dose chemotherapy to ablate hematopoiesis for the eradication of tumor cells and donor cell engraftment. In order to accelerate the recovery of erythropoiesis and reduce the risks associated with erythrocyte transfusion in cancer patients, erythropoietin is used; however, there are no recommendations for its use in patients undergoing HSCT [15,16]. After high-dose chemotherapy followed by HSCT, G-CSF or granulocyte-macrophage colony stimulating factor (GM-CSF) are used to minimize the rates of morbidity and mortality associated with prolonged neutropenia. However, there is no consensus on the optimal use and effectiveness of these drugs in patients with allogeneic transplantation. Despite the decrease in the severity of neutropenia, the rates regarding the duration of hospitalization and mortality in these patients did not decrease, which is associated with a small reduction in the risk of infections [31]. In addition, recent laboratory studies have demonstrated that G-CSF and GM-CSF can also alter T cell and dendritic cell function, which may increase the risk of graft versus host disease (GVHD) [32]. GVHD remains a major complication regarding allo-SCT, affecting up to 40–60% of allo-HSCT patients; it is caused by the activation of donor immunity with the development of cytokine storm. Therefore, immunosuppressants are used to treat and prevent GVHD, including the cytostatic methotrexate (MTX). Although low doses of MTX are used in this category of patients, hematologic toxicity of grades III and IV in allo-HSCT patients are observed in about 40% of individuals, and pancytopenia is one of the most frequent severe toxicities of methotrexate [33,34].

In these cases, the levels of effectiveness of G-CSF or erythropoietin are clearly insufficient; it is necessary to use drugs that can stimulate all hematopoietic germs. Therefore, the search for new effective means of stimulating hematopoiesis suppressed under the influence of chemotherapy is still an urgent task regarding concomitant therapy for cancer patients. Such drugs can be found among the derivatives of sulfated polysaccharides, which, along with low toxicity, have the ability to stimulate hematopoiesis. Recently, fucoidan, isolated from the sea cucumber *Holothuria polii*, was found to increase the recovery of leucocytes and neutrophils in mice after CPh exposure [35]. The authors also noted the tendency for erythrocytes to recover. Previously, we observed similar effects for fucoidan, isolated from the brown seaweed *Chordaria flagelliformis*, and fucosylated chondroitin sulfate, isolated from the sea cucumber *Massinium magnum,* as well as for their modified derivatives [18,21].

Here, we studied the ability of sulfated polysaccharides **CS** and **FCS** isolated from marine animals to stimulate hematopoiesis in CPh-treated mice. The protein r G-CSF was used as a reference. In contrast to the specified growth factor r G-CSF, the studied polysaccharides stimulated all hematopoietic germs, increasing the number of WBCs, RBCs and platelets up to levels comparable to those of the intact control. Moreover, the recovery of the number of WBCs was more balanced in the cases of **CS** and **FCS** application, as an increase in the number of lymphocytes was also observed.

IL-6 is known to play a leading role in the development of cytokine storms and associated organ and multiple organ failure [36]. This cytokine also plays an important role in the pathogenesis of GVHD, and the post-transplant level of IL6 is considered to be a predictive risk factor for severe aGVHD [37,38]. Recent reports have demonstrated that the anti-IL-6 receptor antibody Tocilizumab may reduce the severity of GVHD [39]. However, the effect of Tocilizumab is not durable, and some patients have infections which could be associated with immunosuppressed states. Therefore, the observed decrease in the concentration of IL-6 in the blood of mice after a course of polysaccharides **CS** and **FCS** suggests that compounds of this class can influence the manifestations of allo-HSCT patients.

In vitro studies led to the conclusion that the systemic effect of **CS** and **FCS** was accompanied and possibly mediated by the intensification of CD44-associated cell–cell interactions of bone marrow cells. CD44, as a member of the cell adhesion molecule family, is extensively expressed in bone marrow cells and has previously been reported to play important roles in hematopoietic regulation via CD44–ligand interactions [40,41,42,43,44]. We can assume that **CS** and **FCS** stimulate the expression of CD44 on bone marrow cells and, therefore, mediate the formation of numerous intercellular contacts. As a result, there is a rapid accumulation of cytokines, growth factors and other bioactive molecules in a limited volume of hematopoietic niches, which can lead to an increase in auto- and paracrine activation of proliferation. It is very likely that this effect is the reason for the stimulation of the proliferation of bone marrow cells, mediating an increase in the number of various pools of blood cells in systemic circulation.

Additionally, the studied polysaccharides were shown to be able to restore the proliferative potential of hematopoietic bone marrow stem cells inhibited by chemotherapy drugs. In particular, in the cases of **CS** and **FCS**, an increase in the concentration of the proliferating hematopoietic cells [CD34(+)CD45(+)Ki67(+)] was observed by 8 and 5 times, respectively, after cytostatic agent exposure.

Considering the ability of **CS** and **FCS** to stimulate hematopoiesis and reduce the manifestations of immunosuppression caused by the actions of cyclophosphamide, as well as their ability to reduce the level of IL-6, it seems advisable to study the oligosaccharide derivatives in regard to their potential as promising drugs in complex prophylaxis and immunosuppression therapy.

## 4. Materials and Methods

### 4.1. General Methods

Immunophenotype and membrane-associated markers in bone marrow cells were examined using the anti-mouse antibodies CD34, CD44 and CD45 (Becton Dickinson Bioscience, San Jose, CA, USA). A BD Canto II flow cytometer (Becton Dickinson, San Jose, CA, USA) was used for this study. Sample preparation was carried out in accordance with the manufacturer’s instructions. To evaluate each parameter, the blood of 3 mice of each group was used. Analysis of Ki67(+) cells was performed with a Muse Cell Analyzer (Merck KGaA, Darmstadt, Germany) using the Muse Ki67 Proliferation Kit (EMD Millipore Corporation, Billerica, MA, USA).

The adhesion of bone marrow cells was studied in duplicate using an Agilent xCELLigence real-time cell analysis multiple plates system (RTCA xCELLinge, ACEA Biosciences, Santa Clara, CA, USA). The results were evaluated by assessing the changes of the cell indexes in comparison to the control using the appropriate software.

### 4.2. Sulfated Polysaccharides

The polysaccharide sample **CS** was prepared from a crude extract of *Salmo salar* cartilage via a mild alkaline treatment, followed by anion-exchange chromatography, as described previously [22]. The polysaccharide sample **FCS** was isolated from a body wall of the sea cucumber *Apostichopus japonicus*, as described previously [23].

### 4.3. Animal Model

A total of 30 mice of the Balb/c line (males, weight 20 ± 2 g) were divided into 5 groups of 6 animals each. Before and during the experiment, the animals were placed in standardized vivarium conditions (at 20 ± 2 °C with free access to food and water). To induce myelosuppression, CPh (Endoxan, Baxter, Germany) in a dosage of 100 mg/kg was intraperitoneally injected to animals from 4 of the groups once daily for 4 days. Then, the following sterile solutions (0.2 mL) were administered subcutaneously to all animals for 3 days (once daily): 0.5 mg/mL of **CS** in isotonic sodium chloride solution (group CPh + **CS**), 0.5 mg/mL of **FCS** in isotonic sodium chloride solution (group CPh + **FCS**), 3 nmol/mL of **r G-CSF** (Leucita, Sygardis AqVida, Germany) in isotonic sodium chloride solution (group CPh + **r G-CSF**) and sterile isotonic sodium chloride solution (groups CPh). A sterile isotonic sodium chloride solution was administered to the mice in the control group using the same regime. The animals were euthanized via decapitation after 2 days. Blood of each animal was collected in tubes with ethylenediaminetetraacetic acid (EDTA), the spleen was removed from each animal and smears were imprinted on the polyethylene-coated glasses (Gerhard Menzei GmbH, Thermo Scientific). The fingerprints were fixed in May–Grunwald solution, stained with hematoxylin–eosin (HE) and analyzed via light microscopy. The hematological parameters of the blood were analyzed using an automatic analyzer IDEXX LaserCyte Dx (IDEXX Laboratories, Inc., Westbrook, Maine, USA), determining the concentration of WBCs, platelets and RBCs. Bone marrow cells were isolated from the femurs. Formalin-fixed, paraffin-embedded mouse spleen sections were stained with hematoxylin–eosin (PanEco, Moscow, Russia), analyzed and documented using the light microscopy system (Axioplan 2, Carl Zeiss, Germany).

### 4.4. Cell Model

Bone marrow cells (BM cells) were isolated from the femoral bone of healthy Balb/c mice (male, weight 18 ± 1 g). BM cells were suspended in growth medium based on Dulbecco’s modified Eagle’s medium (DMEM) (Sigma-Aldrich, St. Louis, MO, USA), supplemented with 10% fetal bovine serum (FBS; HyClon, Thermo Fisher, Waltham, MA, USA), 1% penicillin/streptomycin (PanEco, Moscow, Russia) and 4 mM L-glutamine (PanEco, Moscow, Russia) at 37 °C and 5% CO_2_ to a concentration of 520 000 cells/mL. A suspension of BM cells (800 μL) was placed in a 24-well plate (Nunclon, Thermo Fisher, Walthman, MA, USA). Amounts of 100 μL of solutions of **CS** (1 mg/mL), **FCS** (1 mg/mL) and r G-CSF (3 nmol/mL) in isotonic sodium chloride solution or isotonic sodium chloride solution were added to the cells. Additionally, either 100 μL of isotonic sodium chloride solution was added (groups **CS**, **FCS**, r G-CSF, control) or 100 μL of 50 μg/mL cisplatin (groups Cis-**CS**, Cis-**FCS**, Cis-r G-CSF, Cis) was added. After 48 h of incubation, the medium with cells was taken out of the wells, 500 μL of 0.25% trypsin solution (PanEco, Moscow, Russia) was added to each well and the suspension with detached cells was added to the previously collected suspension. The cells were washed with a fresh portion of the medium at 300g for 5 minutes. The sediment was resuspended in the medium. Then, in each group, the level of proliferating cells was assessed after counting the number of cells labeled with Ki67 (Ki67(+) cells), in accordance with the manufacturer’s instructions. The expression of CD44 was assessed in the **CS**, **FCS**, r G-CSF and control groups using the flow cytometry method.

To assess cell adhesion, a suspension of BM cells (100 μL) obtained as described above was incubated in a medium containing 0.1 mg/mL **CS** (**CS** group) or 0.1 mg/mL **FCS** (**FCS** group) in doublets of an E-plate 16 (ACEA Biosciences, San Diego, CA, USA). In the control, 100 μL of growth medium (control group) was added to the cells. The results were examined by assessing the change in the cell index in comparison with the control, which was taken into account in real time using the Agilent xCELLigence real-time cell analysis multiple plates system during incubation for 48 h at 37 °C and 5% CO_2_.

### 4.5. Statistical Analysis

Six animals from each group were used in the in vivo experiments. The in vitro experiments were performed in triplicate. Data from each group are presented in the format of the mean and standard deviation (m ± SD). An analysis of the reliability of the differences was carried out using the *t* criterion. Differences were considered significant at *p* < 0.05.

## 5. Conclusions

In the present study, chondroitin sulfate (**CS**), isolated from cartilage of the fish *Salmo salar*, and fucosylated chondroitin sulfate (**FCS**), isolated from the sea cucumber *Apostichopus japonicus*, showed the ability to stimulate hematopoiesis in a model of cyclophosphamide-induced immunosuppression in mice. Unlike r G-CSF, the studied polysaccharides stimulated an increase in not only the level of white blood cells, but also erythrocytes and platelets due to the effect on progenitor cells of bone marrow. An increase in the proliferative activity of hematopoietic cells CD34(+)CD45(+) was observed after polysaccharide exposure ex vivo. These properties of the sulfated polysaccharides allow us to consider them as potentially promising drugs for the treatment and prevention of the immune status and hematopoietic disorders induced by immunosuppressants during chemotherapy and GVHD treatment.

Ability of the sulfated polysaccharides **CS** and **FCS** to stimulate hematopoiesis expanded the spectrum of their biological activities observed previously [45,46,47,48]. Due to their more regular structures [22,23,49,50,51] compared to those of fucoidans, chondroitin sulfates and fucosylated chondroitin sulfates could be regarded as convenient models for use in the determination of structure–activity relationships and the development of effective hematopoietic stimulators.

## Figures and Tables

**Figure 1 pharmaceuticals-14-01074-f001:**
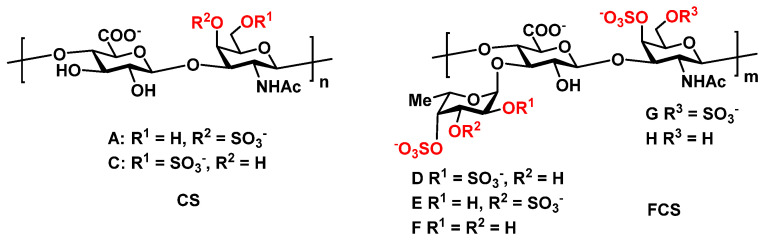
Structures of chondroitin sulfate (**CS**) from the fish *Salmo salar* [22] and fucosylated chondroitin sulfate (**FCS**) from the sea cucumber *Apostichopus japonicus* [23].

**Figure 2 pharmaceuticals-14-01074-f002:**
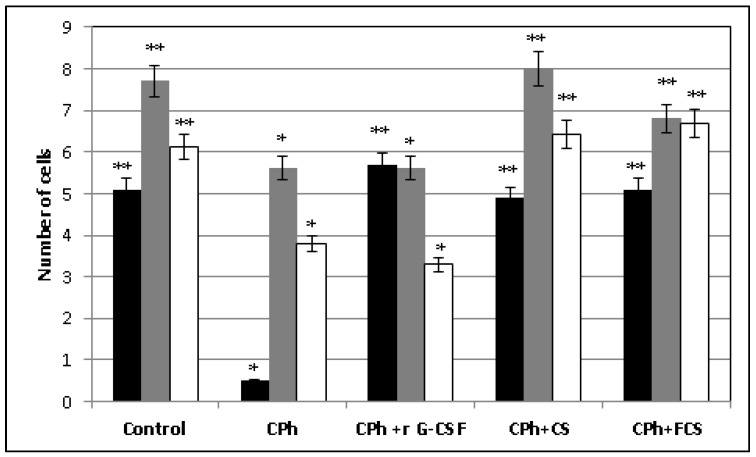
Hematological parameters of mice with cyclophosphamide (CPh)-induced immunosuppression after treatment with recombinant granulocyte colony-stimulating factor (r G-CSF), chondroitin sulfate (**CS**) and fucosylated chondroitin sulfate (**FCS**) (mean ± SD). ■ WBC × 10^6^/mL, ■ RBC × 10^9^/mL, □ platelets × 10^8^/mL. * *p* < 0.05 vs. control, ** *p* < 0.05 vs. CPh.

**Figure 3 pharmaceuticals-14-01074-f003:**
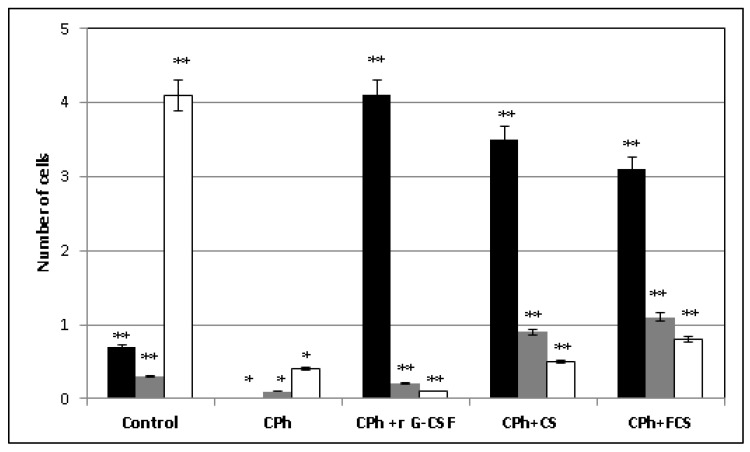
Different populations of leucocytes (WBCs) in blood of mice with CPh-induced immunosuppression after treatment with recombinant granulocyte colony-stimulating factor (r G-CSF), chondroitin sulfate (**CS**) and fucosylated chondroitin sulfate (**FCS**) (mean ± SD). Neutrophils × 10^6^/mL, ■ monocytes × 10^6^/mL, □ lymphocytes × 10^6^/mL. * *p* < 0.05 vs. control, ** *p* < 0.05 vs. CPh.

**Figure 4 pharmaceuticals-14-01074-f004:**
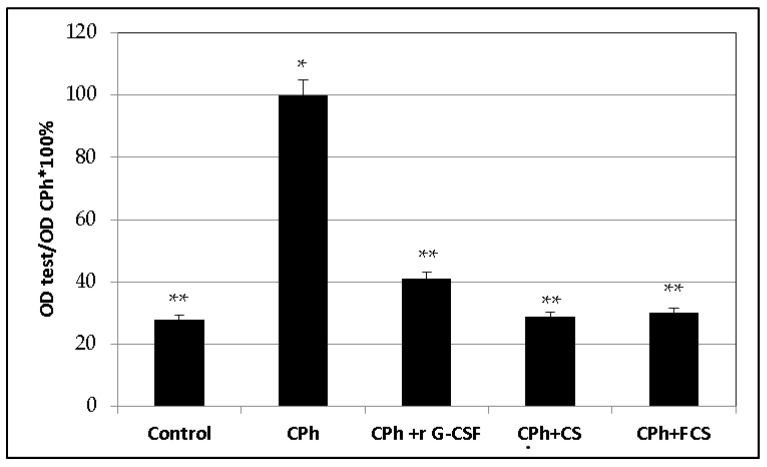
The level of IL-6 in blood of mice with CPh-induced immunosuppression after treatment with recombinant granulocyte colony-stimulating factor (r G-CSF), chondroitin sulfate (**CS**) and fucosylated chondroitin sulfate (**FCS**) (mean ± SD). * *p* < 0.05 vs. control, ** *p* < 0.05 vs. CPh.

**Figure 5 pharmaceuticals-14-01074-f005:**
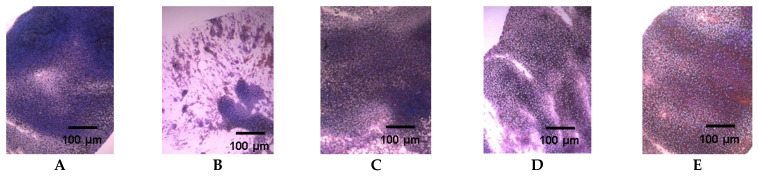
Spleen fingerprints of mice with CPh-induced immunosuppression after treatment with tested compounds (hematoxylin–eosin staining). (**A**) Control, (**B**) CPh, (**C**) CPh-r G-CSF, (**D**) CPh-**CS**, (**E**) CPh-**FCS**. Original magnification ×400.

**Figure 6 pharmaceuticals-14-01074-f006:**
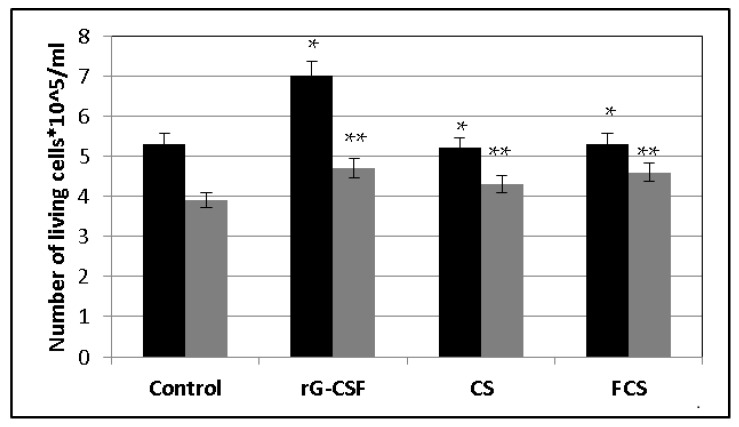
Number of bone marrow living cells after the treatment with recombinant granulocyte colony-stimulating factor (r G-CSF), chondroitin sulfate (**CS**) and fucosylated chondroitin sulfate (**FCS**) (mean ± SD). ■ Bone marrow cells, ■ bone marrow cells after cisplatin exposure. * *p* < 0.05 vs. control, ** *p* < 0.05 vs. Cis.

**Figure 7 pharmaceuticals-14-01074-f007:**
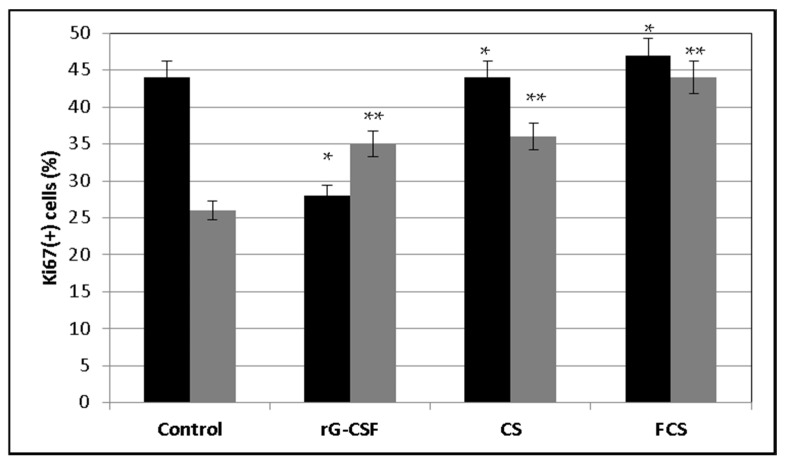
Concentration of Ki67(+) cells (%) after treatment with recombinant granulocyte colony-stimulating factor (r G-CSF), chondroitin sulfate (**CS**) and fucosylated chondroitin sulfate (**FCS**) (mean ± SD, *p* < 0.05). ■ Bone marrow cells, ■ bone marrow cells after cisplatin exposure. * *p* < 0.05 vs. control, ** *p* < 0.05 vs. Cis.

**Figure 8 pharmaceuticals-14-01074-f008:**
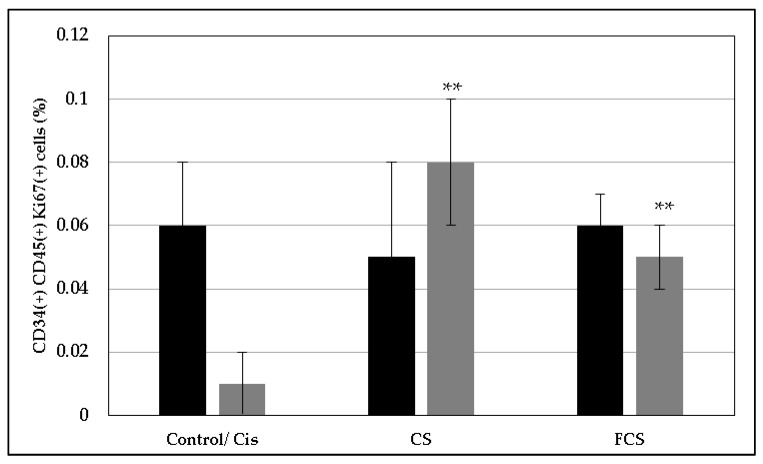
Concentration of CD34(+)CD45(+)Ki67(+) cells (%) after the treatment with chondroitin sulfate (**CS**) and fucosylated chondroitin sulfate (**FCS**) (mean ± SD). ■ Bone marrow cells, ■ bone marrow cells after cisplatin exposure. ** *p* < 0.05 vs. Cis.

**Figure 9 pharmaceuticals-14-01074-f009:**
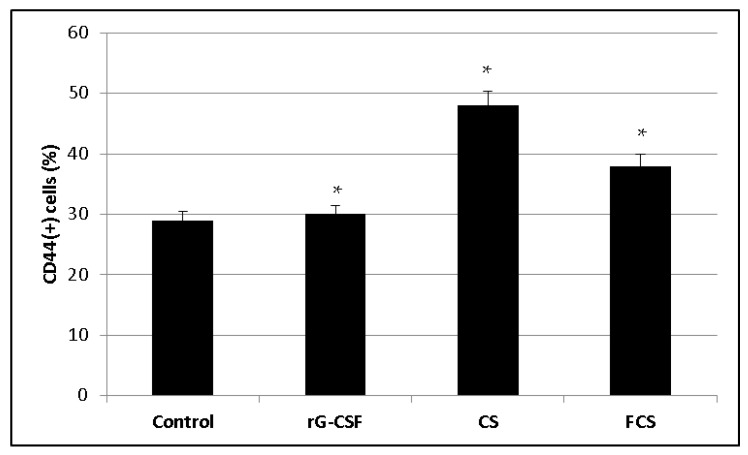
Level of CD44(+) nuclear bone marrow cells (%) after treatment with recombinant granulocyte colony-stimulating factor (r G-CSF), chondroitin sulfate (**CS**) and fucosylated chondroitin sulfate (**FCS**) (mean ± SD, *p* < 0.05). * *p* < 0.05 vs. control.

**Figure 10 pharmaceuticals-14-01074-f010:**
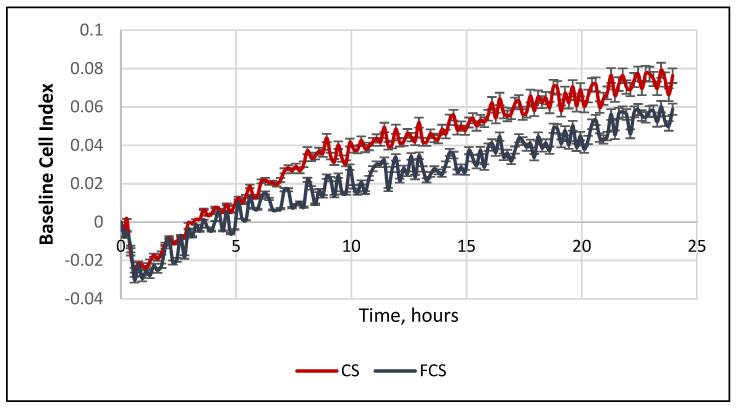
The effect of **CS** and **FCS** on the adhesion of mouse bone marrow cells (red—**CS**, blue—**FCS**). The cell index of the **CS** and **FCS** groups is normalized to the control values.

## Data Availability

Data is contained within the article and Appendix A.

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
