# Peer review of "Chondroitin Sulfate and Fucosylated Chondroitin Sulfate as Stimulators of Hematopoiesis in Cyclophosphamide-Induced Mice"

_pharmaceuticals, 2021, doi:10.3390/ph14111074_

Round 1
Reviewer 1 Report
Authors revised their manuscript. The following issues still present and the improvement of these problems may need long while. 1) REgarding Fig. 1, atr least for this manuscript, g-SCF only, CS only and FCS only groups should be set. I am wondering whether or not the dosage of CS/FCS are toxic or not? 2) Whole figures,,,, now figures possessed "*", and "**". However, it is not possible to know which and which were compared. In addition, did authors try one-way ANOVA? 3) Regarding Fig, 4, authors still showed IL-6 as a marker for hepatotoxicity. How about enzymes such as AST, ALT and γGT? How about histology? 4) What kind of cell type(s), for example, the progenitor of granulo/macrophage lineage, erythrocyte lineage, revealed the increase of CD44? 5) Regarding Fig. 11, the experimental conditions as well as how did authors take photos would be documented? 6) All the data seemed to be just phenomenon. We need some mechanistic analyses. Do authors hope to emphasize that CS/FCS show colony-stimulating activity (especially for granulocyte/macrophage lineage) more than standard G-CSF via CD44? If so, authors should try neutralizing CD44 by antibody, or siRNA or other methods, Comprehensively, this manuscript is still immature and I feel authors should re-challenge the experiments to observe some mechanistic approaches.Author Response
1) Regarding Fig. 1, at least for this manuscript, g-SCF only, CS only and FCS only groups should be set. I am wondering whether or not the dosage of CS/FCS are toxic or not?
Authors: The aim of the study presented in the manuscript was the elucidation of general effect of selected sulfated polysaccharides on hematopoiesis in immunosuppressed mice as on model of GVHD. Detailed investigation of the mechanism of the polysaccharide action including recommended experiments (g-SCF only, CS only and FCS only groups) was out of the frame of present study. But obtained promising results made reasonable of mentioned studies on next steps. The studied polysaccharides are non-toxic at studied concentrations, that were shown previously in our papers cited in the manuscript (for an example see reference 22).
2) Now figures possessed "*", and "**". However, it is not possible to know which and which were compared. In addition, did authors try one-way ANOVA?
Authors: In the legend to the figures “*” is pointed as the statistically significant difference to the control group, and “**” is pointed as the statistically significant difference to the CPh group. An analysis of the reliability of the differences was carried out using the t test. It is known that the one-way ANOVA is used to test for differences among at least three groups, since the two-group case can be covered by a t-test. This is a usual practice. Respected notes were included into legends to each Figure and are present in present version.
3) Regarding Fig, 4, authors still showed IL-6 as a marker for hepatotoxicity. How about enzymes such as AST, ALT and γGT? How about histology?
Authors: Hepatotoxic action of cytostatic drugs is well known. Also it was shown, that an increase in the serum level of IL-6 is connected with damage of the liver parenchyma (for example, see Ref. 8). As the elucidation of the hepatotoxic effect of the polysaccharides was not the aim of this work, we have only referred to the published observations to explain the increasing of the level of IL-6 in cyclophosphamide-induced mice. From our point of view, in this context, the ability of the studied polysaccharides to reduce the level of IL-6, which plays an important role in the pathogenesis of GVHD, is more important. Along with the ability to stimulate hematopoiesis, it could have clinical significance in complex therapy of allo-HSCT patients.
4) What kind of cell type(s), for example, the progenitor of granulo/macrophage lineage, erythrocyte lineage, revealed the increase of CD44?
Authors: In this series of experiments, the expression of CD44 by nuclear bone marrow cells was studied. From our point of view, the stimulation of all hematopoietic germs that we found under the action of the polysaccharides accompanied by an increase in the expression of CD44 by bone marrow cells confirms the stimulation of hematopoiesis inhibited by cyclophosphamide in mice.
5) Regarding Fig. 11, the experimental conditions as well as how did authors take photos would be documented?
Authors: The photographs presented in the manuscript demonstrate the differences in the cellular composition of the red and white pulps of the spleen of mice in different groups. In this case we used very well established approach (doi.org/10.17504/protocols.io.ihxcb7n). It is known that the white pulp is the site of the concentration of lymphocytes (mainly T cells), while the red pulp consists mainly of RBCs, along with platelets and neutrophils. Thus, the picture of desolation of the spleen pulp (decrease in the number of spleenocytes) reflects the degree of inhibition of hematopoiesis in the animal and a decrease in the immune potential (immunosuppression) due to leukopenia. Thus, the replenishment of the cellular composition of the spleen indicates the restoration of hematopoiesis. We used standard histological methods of sampling, fixation and staining of samples described in Section 4.3. The legends to Figure 5 and 1S were corrected.
6) All the data seemed to be just phenomenon. We need some mechanistic analyses. Do authors hope to emphasize that CS/FCS show colony-stimulating activity (especially for granulocyte/macrophage lineage) more than standard G-CSF via CD44? If so, authors should try neutralizing CD44 by antibody, or siRNA or other methods.
Authors: The ability of certain sulfated polysaccharides to stimulate the hematopoiesis in fact was discovered previously (see cited refs. 18, 21, 35). The present manuscript was prepared specially for the issue of this journal dedicated to GVHD and not to mechanistic studies of hematopoiesis including recommended experiments (neutralizing CD44 by antibody). Nevertheless we described in the manuscript the results of performed experiments with CD34(+) CD45(+) bone marrow steam cells which demonstrated significant increase of their proliferation (increase of Ki67(+) cells) after the treatment with studied compounds. It is an important step for deeper investigation of hematopoiesis mechanism. All conclusions of this work are based on the obtained experimental data which were confirmed by modern methods of statistical processing. The main objective of the study was to evaluate the ability of the tested substances to stimulate hematopoiesis. Using various methodological approaches (hematological, morphological, flow cytometry and immunofluorescence microscopy), data on the reliable activity of CS and FCS were obtained. Thus we established not only the ability of CS and FCS to stimulate various hematopoietic links in mice with depression of hematopoiesis induced by cyclophosphamide. We also showed an increase in the proliferative activity of hematopoietic stem cells (CD34(+) CD45(+)) by expression of the proliferation marker Ki67 and enhanced formation of cell clusters. In addition, an increase in the expression of the CD44 marker, which plays an important role in the CD44-associated cell-cell interactions of bone marrow cells, was demonstrated by hematopoietic cells in the bone marrow. CD44 as a member of the cell adhesion molecule family is extensively expressed by bone marrow cells and has been previously reported to play important roles in hematopoietic regulation via CD44-ligand interactions. The results obtained suggest that CS and FCS can be considered as potentially promising drugs for concomitant therapy of cancer patients and, in particular, for stimulating hematopoiesis after stem cell transplantation including the GVHD.
Overall, we thank the reviewer for all comments and hope for acceptance of our responses.
Reviewer 2 Report
Authors have mainly changed the text in the discussion section with respect to the previous form. I feel that this manuscript still contains flaws in study design, unsupported conclusions and technical limits. In general, it is highly descriptive and the conclusions are too speculative considering the data presented.
Author Response
The ability of certain sulfated polysaccharides to stimulate the hematopoiesis in fact was discovered previously (see cited refs. 18, 21, 35). The present manuscript was prepared specially for the issue of this journal dedicated to GVHD and not to mechanistic studies of hematopoiesis including recommended experiments (neutralizing CD44 by antibody). Nevertheless we described in the manuscript the results of performed experiments with CD34(+) CD45(+) bone marrow steam cells which demonstrated significant increase of their proliferation (increase of Ki67(+) cells) after the treatment with studied compounds. It is an important step for deeper investigation of hematopoiesis mechanism. All conclusions of this work are based on the obtained experimental data which were confirmed by modern methods of statistical processing. The main objective of the study was to evaluate the ability of the tested substances to stimulate hematopoiesis. Using various methodological approaches (hematological, morphological, flow cytometry and immunofluorescence microscopy), data on the reliable activity of CS and FCS were obtained. Thus we established not only the ability of CS and FCS to stimulate various hematopoietic links in mice with depression of hematopoiesis induced by cyclophosphamide. We also showed an increase in the proliferative activity of hematopoietic stem cells (CD34(+) CD45(+)) by expression of the proliferation marker Ki67 and enhanced formation of cell clusters. In addition, an increase in the expression of the CD44 marker, which plays an important role in the CD44-associated cell-cell interactions of bone marrow cells, was demonstrated by hematopoietic cells in the bone marrow. CD44 as a member of the cell adhesion molecule family is extensively expressed by bone marrow cells and has been previously reported to play important roles in hematopoietic regulation via CD44-ligand interactions. The results obtained suggest that CS and FCS can be considered as potentially promising drugs for concomitant therapy of cancer patients and, in particular, for stimulating hematopoiesis after stem cell transplantation including the GVHD.
Overall, we thank the reviewer for all comments and hope for acceptance of our responses.
Reviewer 3 Report
The authors have adequately revised the manuscript in order to address the points raised by the reviewers. This revised manuscript can be accepted for publication.
Author Response
We thank the Reviewer 3 for evaluation of the manuscript and endorsement of its acceptance.
Reviewer 4 Report
- What is the mechanism for sulfated polysaccharides CS and FCS giving the activity as hematopoietic stimulators?
- Once CS and FCS are administered subcutaneously to animals, what is the clearance pathway?
- What was the purity of CS and FCS? Was there any endotoxin in the product?
Author Response
- What is the mechanism for sulfated polysaccharides CS and FCS giving the activity as hematopoietic stimulators?
Authors: The ability of certain sulfated polysaccharides to stimulate the hematopoiesis in fact was discovered previously (see cited refs. 18, 21, 35). The present manuscript was prepared specially for the issue of this journal dedicated to GVHD and not to mechanistic studies of hematopoiesis including recommended experiments (neutralizing CD44 by antibody). Nevertheless we described in the manuscript the results of performed experiments with CD34(+) CD45(+) bone marrow steam cells which demonstrated significant increase of their proliferation (increase of Ki67(+) cells) after the treatment with studied compounds. It is an important step for deeper investigation of hematopoiesis mechanism. All conclusions of this work are based on the obtained experimental data which were confirmed by modern methods of statistical processing. The main objective of the study was to evaluate the ability of the tested substances to stimulate hematopoiesis. Using various methodological approaches (hematological, morphological, flow cytometry and immunofluorescence microscopy), data on the reliable activity of CS and FCS were obtained. Thus we established not only the ability of CS and FCS to stimulate various hematopoietic links in mice with depression of hematopoiesis induced by cyclophosphamide. We also showed an increase in the proliferative activity of hematopoietic stem cells (CD34(+) CD45(+)) by expression of the proliferation marker Ki67 and enhanced formation of cell clusters. In addition, an increase in the expression of the CD44 marker, which plays an important role in the CD44-associated cell-cell interactions of bone marrow cells, was demonstrated by hematopoietic cells in the bone marrow. CD44 as a member of the cell adhesion molecule family is extensively expressed by bone marrow cells and has been previously reported to play important roles in hematopoietic regulation via CD44-ligand interactions. The results obtained suggest that CS and FCS can be considered as potentially promising drugs for concomitant therapy of cancer patients and, in particular, for stimulating hematopoiesis after stem cell transplantation including the GVHD.
- Once CS and FCS are administered subcutaneously to animals, what is the clearance pathway?
Authors: Since we used Filgrastim as a reference drug, which is used in clinical practice under conditions of subcutaneous administration, this route of administration was also used for the tested substances. It is obvious that water-soluble CS and FCS, as well as Filgrastim, enter the systemic circulation and the bone marrow.
- What was the purity of CS and FCS? Was there any endotoxin in the product?
Authors: The purification of the polysaccharides was performed using ion-exchanched and gel-exclusive chromatography methods described in References 22 and 23. The purity of the compounds was confirmed by exclusive chromatography and the NMR spectroscopy methods. Therefore there is no reason to suspect the endotoxin contamination of the test compounds.
Overall, we thank the reviewer for all comments and hope for acceptance of our responses.
Round 2
Reviewer 1 Report
Authors wrote as ”the studied polysaccharides were hown to stimulate release of white and red blood cells, as well as platelets from bone marrow in immunosuppressed mice". However, no direct evidences of this description in this manuscript. Much more simple results such as peripheral WBC, RBC and platelets counts should be shown, if authors emphasize this point. In addition, in vivo results should be shown, for example, survival of mice, liver pathology and others. We could pick up Fig. 11, but, it should be differed as hematopoietic stem cell colony and other differentiated colonies.
Author Response
Reviewer 1: Authors wrote as ”the studied polysaccharides were shown to stimulate release of white and red blood cells, as well as platelets from bone marrow in immunosuppressed mice". However, no direct evidences of this description in this manuscript. Much more simple results such as peripheral WBC, RBC and platelets counts should be shown, if authors emphasize this point.
Authors: The peripheral WBC, RBC and platelets counts are shown on Figure 2.
Reviewer 1: In addition, in vivo results should be shown, for example, survival of mice, liver pathology and others.
Authors: In vivo results, such as peripheral WBC, RBC and platelets counts, spleen fingerprints and spleen sections are now presented on Figures 2, 5 and 1S, respectively.
Reviewer 1: We could pick up Fig. 11, but, it should be differed as hematopoietic stem cell colony and other differentiated colonies.
Authors: Following to this recommendation we have moved Fig. 11 to supplementary (now it is Fig. 2S), as it was just illustration of the formation of cell clusters measured using a RTCA ACEA cell analyzer.
Reviewer 2 Report
This revision makes few changes from the original. Further, technical weaknesses are still present. The additional experiment included to show "An increase in the proliferative activity of hematopoietic stem cells CD34(+)CD45(+) after the polysaccharide exposure ex vivo" has many flaws in its experimental design. Mouse stem cells can be identified by means of complex immunophenotyping strategies. CD45 and CD34 do not identify hematopoietic stem cells in the mouse bone marrow.
The text has many basic grammatical errors and odd word usages. Page 4: text is in Russian.
Author Response
Reviewer 2: This revision makes few changes from the original. Further, technical weaknesses are still present.
Authors: Following to this recommendation we have improved stylistically the manuscript which is transformed now into “Communication”. Also we have moved Fig. 11 to supplementary and have modified the comments to Figure 10. The manuscript was also subjected to English editing (see below).
Reviewer 2: The additional experiment included to show "An increase in the proliferative activity of hematopoietic stem cells CD34(+)CD45(+) after the polysaccharide exposure ex vivo" has many flaws in its experimental design. Mouse stem cells can be identified by means of complex immunophenotyping strategies. CD45 and CD34 do not identify hematopoietic stem cells in the mouse bone marrow.
Authors: It is known that the immunophenotype of stem cells is quite complex, unstable and can be characterized by a wide range of markers, depending on many factors. In our work we used the antibody to mouse CD45 (it is one of the most abundant hematopoietic markers) together with the antibody to mouse CD34 which is the adhesion molecule with a role in early hematopoiesis by mediating the attachment of stem cells to the bone marrow extracellular matrix or directly to stromal cells. Therefore, increasing of the proliferation in the population CD34(+)CD45(+) cells could be considered as a confirmation of hematopoiesis stimulation.
Reviewer 2: The text has many basic grammatical errors and odd word usages.
Authors: Following to this recommendation the manuscript was subjected to editing using the proposed service of MDPI (via https://www.mdpi.com/authors/english; Payment confirmation for English editing invoice: english-35707).
Round 3
Reviewer 1 Report
Author modified their manuscript.
Author Response
We thank the reviewer for the evaluation of the manuscript.
Reviewer 2 Report
The authors have answered all my comments to the best of their ability. I would only remove the definition of "stem cells" and define CD34/CD45 double positive cells as bone marrow hematopoietic cells.Author Response
We thank the Reviewer for the evaluation of the manuscript.
As recommended, we have remove the definition of "stem cells" and define CD34/CD45 double positive cells as bone marrow hematopoietic cells. The corrections have been made in Abstract, Results, Discussion and Conclusion.